# Interventions supporting the translation of gerontological evidence into practice to optimize functional outcomes for hospitalized older adults: A scoping review

Kathleen F. Hunter[1]*, Jeffrey I. Butler[1], Shovana Shrestha[1], C. Allyson Jones[2], Adrian Wagg[3], Nick Millar[1], Frances Carr[3], Sherry Dahlke[1]

1 Faculty of Nursing, University of Alberta, Edmonton, Alberta, Canada, 2 Department of Physical Therapy, University of Alberta, Edmonton, Alberta, Canada, 3 Division of Geriatric Medicine, Department of Medicine, University of Alberta, Edmonton, Canada

* kathleen.hunter@ualberta.ca

## Abstract

### Background and objectives

Hospitalized older adults are at risk for both physical and cognitive functional decline. To support the implementation of interventions optimizing their function, both health-care professional knowledge and other factors that may impact practice change should be addressed. The objective of this study was to examine the characteristics and type of interventions supporting the translation of gerontological evidence into practice to optimize functional outcomes for hospitalized older adults.

### Materials and methods

This scoping review used the guidelines recommended by Peters et al. Six electronic databases were searched from database inception to 2024. Included studies were primary research studies that 1) described an educational intervention for healthcare professionals as part of an initiative to promote practice change and 2) assessed functional outcomes (physical and/or cognitive),

### Results

Thirty-eight studies were deemed eligible. Numerous interventions to support implementation of knowledge to improve functional outcomes were identified. In addition to educational strategies, other interventions addressed care processes, changes to the built environment, administration/management support, leadership support/facilitation, and discharge/transition planning. Studies employed a range of measures to assess functional outcomes among hospitalized older adults.

**Data availability statement:** All relevant data are within the manuscript and its Supporting Information files.

**Funding:** KFH was awarded a Knowledge Synthesis Grant from the Seniors Health Strategic Clinical Network™ (SCN), Alberta Health Services (no grant # assigned). Alberta Health Services did not play any role in the the study design, data collection and analysis, decision to publish, or preparation of the manuscript. https://www.albertahealthservices.ca/research/Page14021.aspx.

**Competing interests:** The authors have declared that no competing interests exist.

## Discussion

Older studies tended to focus on nurse-driven interventions, and newer ones on interprofessional interventions. Knowledge gaps were highlighted in terms of theory, research designs, intervention descriptions, country representation, policy, environmental changes, and leadership roles. Future evaluations of interventions to enhance older adults' functioning should examine effectiveness and compare single- and multi-intervention programs. Strategies based on implementation science theory are needed to ensure successful uptake of effective interventions, while also addressing real-world issues in hospital units, such as short staffing, resource constraints, and overcrowding.

## Introduction

Older adults in hospital are at risk for both physical and cognitive decline [1–3].

Physical activity in acutely hospitalized older adults is highly limited [4,5], and the loss of ability to perform activities of daily living (ADL) is a common problem. Almost one third of this patient population experience functional decline at discharge [6]. Crucially, if older adults do not return to their pre-hospitalization level of function within 30 days of discharge, any functional decline is likely to become permanent [7].

Many older adults have highly complex care needs and entering the hospital system, which prioritizes acute treatment and system efficiency [8], can result in functional losses and poor outcomes [9,10]. Factors including limited mobility and suboptimal continence care are associated with functional decline in hospital and can contribute to further decline post-discharge [10,11]. Consequently, optimizing function for this patient population should be a health system priority. To address this concern, interventions impacting practice change should be addressed along with healthcare professionals' knowledge [12]. Informed by an ecological lens from implementation science [13] we conceptualize intervention implementation as influenced by determinants at multiple levels (micro, meso, and macro), which must be addressed conjointly. This scoping review sought to synthesize interventions to support implementation of evidence from gerontological education for hospital-based healthcare professionals and/or interprofessional teams to maintain and restore older adults' functioning.

In acute care settings, healthcare professionals endeavour to maintain older patients' pre-hospitalization baseline function, to avoid functional losses becoming irreversible. To optimize function and limit preventable hospital-acquired functional loss, there is a need to shift the focus of care for hospitalized older people to the preservation and restoration of function [14,15]. For example, evidence suggests that engaging hospitalized older persons in mobility [16] and basic activities of daily living (e.g., toileting, dressing) [17] may prevent both physical and cognitive decline. The implementation of function-oriented approaches, however, has been hampered by inadequate gerontological knowledge on the part of healthcare professionals, which explains our focus on education as a starting point in this review [18–21].

Yet, education alone does not change practice and not all gerontological education initiatives will spur a transformation in older adults' care [22]; a wide variety of factors (e.g., dedicated champions, strong communication, effective teamwork, and appropriate resources) must be recognized as integral to promoting knowledge uptake [23,24]. Consequently, we conceptualize healthcare professional education as necessary to catalyze a shift to more function-oriented approaches, but as insufficient in isolation. It is thus essential to develop a broader understanding of both the education that has been employed, as well as other interventions that have been implemented in acute hospital settings to support the application of new knowledge to optimize older adults' functioning.

Previous studies have underscored that, in acute care settings, a wide array of micro, meso and macro factors (e.g., patient attitudes, organizational culture/climates, financial resources, leadership) [13] shape healthcare professionals' perceptions around the feasibility of implementing new approaches to care [25]. However, it remains unknown what educational interventions have been employed, to promote function-oriented care for acutely hospitalized older adults.

## Objective

Our objective was to determine what educational interventions for hospital-based healthcare professionals and/or interprofessional teams, have been implemented to optimize older adults' functional outcomes in hospital settings. The objective was identified as a research priority in a provincial priority-setting partnership [26]. As we could not locate any existing review in the hospital setting, we determined that a broad scoping review to map the breadth of the available evidence was most appropriate [27,28]. Accordingly, we mapped the evidence on educational interventions for healthcare professionals that have been implemented as part of an initiative to drive practice change.

## Materials and methods

We used the guidelines for scoping reviews recommended by Peters et al. [29]. According to Munn et al. [28], scoping reviews are "a type of evidence synthesis that aim to systematically identify and map the breadth of evidence available on a particular topic, field, concept, or issue, often irrespective of source (i.e., primary research, reviews, non-empirical evidence) within or across particular contexts" (page 1). Guided by this approach, we formulated the research question; identified relevant studies; selected studies; charted/extracted data; and collated, summarized, and reported results. The PRISMA-ScR (Preferred Reporting Items for Systematic Reviews and Meta-Analyses: Extension for Scoping Reviews) guided our reporting [29,30] (See S1 Table).

### Inclusion/exclusion criteria

We included primary studies of any design, with no restrictions on language/year of publication. Eligible sources that were not published in English were translated using Google Translate. Studies focused on initiatives or care delivery models to improve functional outcomes for adults aged 65+ (in line with the age cutoff used in many World Health Organization reports [31]) in acute care hospital settings. Eligible studies 1) described educational interventions for healthcare professionals as a part of an initiative to promote practice change (e.g., care process change, environmental modifications), 2) assessed functional outcomes in older adults (physical or cognitive), and 3) addressed single or multiple interventions. We excluded reviews and editorials, non-hospital studies, studies focused on adults younger than 65 years old, and studies that did not describe an educational intervention component (see S2 Table).

### Search strategy

An academic health sciences librarian developed the search strategy and conducted the search for relevant published studies. Searches were conducted in 2021 and updated in 2023. Search terms included a combination of natural language vocabulary and controlled terms (subject headings), where available. Natural language terms were derived from three main concepts: 1) educational interventions, 2) older hospitalized patients, and 3) Basic Activities of Daily Living

(BADL) interventions, including interventions addressing specific elements such as mobility, ambulation, and personal hygiene. Search terms for BADL also included language for self-care, functional status, and functional decline (see S3 Table for search strategies for MEDLINE). The search focused on physical function and BADL, but both physical and cognitive outcomes were abstracted from any included studies. Both physical and cognitive outcomes were abstracted because evidence indicates that attending to physical function and BADL can also support cognition. Accordingly, if cognitive function was measured and reported, we included it in our abstraction to lend context to the physical domains that were measured. We conducted the search in six electronic databases: MEDLINE via OVID (1946 – Present); EMBASE via OVID (1974 – Present); Cumulative Index for Nursing and Allied Health Literature (CINAHL) via EBSCOhost (1946 – Present); Scopus via Elsevier (1972 – Present); Cochrane Library via Wiley (1992 – Present); TRIP Pro Medical Database (1997 – Present). We additionally conducted hand searches of the reference lists of included studies.

### Study selection

Identified studies were uploaded to Covidence to support knowledge synthesis. After removal of duplicates, research team member pairs (SS and JIB; KFH and SD) independently screened the abstracts. The full text versions of potentially relevant studies were reviewed as pairs and any discrepancies were resolved through discussion and involvement of a third reviewer, if consensus could not be achieved through discussion. Interrater agreement was not evaluated but regular consensus meetings were held for abstract screening.

### Data extraction

Research assistants (NM and SS) conducted data extraction, which was verified by the principal investigator (KFH) and research coordinator (JIB). We extracted the following elements: (a) study characteristics: author, year, country; (b) study objectives, design, setting, sample; (c) education component(s), contextual component(s) (e.g., level of intervention, care/ team process, facilitation) and functional outcomes (see S4 Table) using a standardized extraction form.

### Data analysis

We employed an inductive approach that facilitated identification of unanticipated patterns in the data, as well as an ecological lens that allowed us to differentiate between micro, meso, and macro levels of healthcare practice change [13,32]. A descriptive numerical summary of the included studies (overall number of studies included, types of study designs, year of publications, countries where studies were conducted, and study settings) and narrative summary of how review findings address the study aim and research question were developed [29], and key findings and gaps in the research evidence were identified. We developed preliminary categories, which were iteratively refined as analysis progressed, to provide a descriptive summary of the findings reported in the studies. Given that scoping reviews should not address targeted questions of intervention feasibility or effectiveness [33], we used our descriptive summary to map the educational interventions/practice change initiatives, their purposes, the division of interprofessional roles, as well as the outcomes examined and how they were measured.

## Results

Database searches located 8,702 citations, 299 of which were included in full-text screening (see S1 Fig for PRISMA Flow Diagram) [30]. Of the 38 Included studies which were conducted between 1993 and 2023; 29 were non-randomized studies with a control [34−62] five were randomized controlled trials (RCTs) [63–67]; three were mixed-method studies [68–70], and one was a quantitative descriptive study [71]. Most studies were conducted in the United States (n = 10), Australia (n = 6), Canada (n = 4) and Israel (n = 3), two in Italy, Netherlands, UK, Germany, and Spain, and one in Brazil, Belgium, Portugal, Thailand, and China. The sample size of the included studies ranged from 19 to 12,490 older adults (see S4 Table for a description of the included studies).

## Geriatric/gerontological education initiatives

More than half the studies (n = 23) reported a single educational initiative [34,47,48 38,39,41,44,46,49,53,54,56,58,59,61–63,65,67–69,71], while the remaining studies reported two or more educational initiatives [35–37,41,43,45,51,52,55,57,60,64,66,70].

The educational initiatives were described as training (n = 22 studies), education session (n = 12), lecture/tutorial (n = 5), in-service (n = 4), poster (n = 4), education module (n = 3), as well as workshop, on the job coaching/training, presentation, pocket cards, resources, and material (n = 2), and seminar, booklet, videos, handout, quiz, case vignette (n = 1) (see Table 1 for frequency of educational strategies reported). Half of the studies targeted a single clinical outcome (n = 19) [38,42,45,48–50,53,54,56,57,59–61,63,67–71], and half (n = 19) addressed multiple clinical outcomes [34–37,39,41,43,44,46,47,51,52,55,58,62,64–66]. Most single-outcome education focused on mobility (n = 11), followed by delirium (n = 7) and cognitive deficits (n = 1). Studies focusing on multiple clinical outcomes examined diverse topics such as: dementia, communication with patients, falls, and incontinence. Mobility and delirium were common clinical outcomes in studies targeting multiple outcomes (see S4 Table). Education targeted nurses (n = 18) [38,40,43,45–49,51–53,55,59,62,63,66,68,71] and various members of the interprofessional team (n = 20) [34–36,39,41,44,50,52,54,56–58,60,61,64,65,67,69,71].

## Interventions to support implementation of gerontological knowledge

We developed the following categories to characterize the interventions that have been implemented to support the implementation of gerontological knowledge and promote practice change: intervention level, care process changes, changes to the built environment, administration/management support, leadership support/facilitation, and discharge/transition planning. See Table 2. Of note, 16 studies reported on interprofessional interventions, with 16 relying on nurses to implement the intervention.

**Intervention level.** Interventions were categorized into three levels based on scale, with small-scale defined as a single care unit focused on one or two geriatric concerns, medium scale defined as one or more care units addressing multiple geriatric concerns, and large-scale defined as multiple hospitals addressing multiple geriatric concerns. There were four large-scale interventions, one included four hospitals working with regional health organizations in a formalized collaboration [39]. The other three involved scale up/spread of two previously tested care models: the hospital elder life program (HELP) to 13 hospitals [71], and the EAT, WALK, ENGAGE program to multiple units in four hospitals [64,65]. Sixteen were medium-scale interventions that had multifocal geriatric protocols in one hospital and 12 were small-scale, focused on one intervention on one unit.

Medium-scale interventions focused on a variety of interventions that included combinations directed at cognition, pain, continence, mobility, nutrition/hydration, sleep and pressure ulcers, physical function/falls, family involvement, medication reviews/protocols, geriatric/frailty screening, and volunteers [34–37,40,43–48,51,53–55,58–60,62,66,67]. Of these, eleven involved nurses implementing interventions [34,45–48,51,52,55,59,62,66], and the rest were interprofessional.

Large-scale studies focused on similar interventions to the medium-scale studies but were implemented in multiple hospitals [39,64,66,71]. All involved an interprofessional approach.

**Care process changes.** Care process changes ranged from a single focused intervention to models of care that addressed multiple geriatric issues. Small-scale studies focused on mobility [38,41,56,61,63,68–70], cognition [50,52], falls [57], or continence and mobility [49,68]. Of these, five focused on nurses implementing interventions [38,42,49,63,68], while the remainder involved an interprofessional approach.

**Changes to the built environment.** Environmental changes included walking trails [43,44,47,64–66,68,70], orientation strategies (e.g., clocks) [40,44,45,47,51–53,59,60,64–66,68], modifications/decluttering [38,40,48,52,58,61,62,66,71], and sensory and mobility aids [57,59,65]. One study examined the impact of adjusted seating and bed heights [47].

**Table 1. Details on Focus of Educational Strategies Utilized by Studies.**

| S.N | First author (Year) | HCPs included in education | Topics covered. (e.g., delirium, mobility) | Education strategies used (e.g., in-service, workshops, pockets cards) |
|---|---|---|---|---|
| 1. | Allegri et al. (2022) | Yes: Staff (doctors, nurses, physical therapists and healthcare assistants). | Dementia, cognitive impairment, depression bipolar psychotic disorder, delirium, pain, communication and interaction with patients | Training using teaching modules. |
| 2. | Apolinario et al. (2022) | Yes (nurses, physiotherapist, speech therapist, nutritionist, psychologist, and clinical pharmacologist). | Fall, pressure injury, bronchoaspiration, delirium, mechanical restraint, malnutrition, medication iatrogenesis, and functional decline. | Education targeted to train professionals from different areas. Staff were given resources such as booklets, video lessons, infographics, on-site small group training, and training workshops. |
| 3. | Bakker et al. (2014). | Yes (Nurses and Physicians). | Delirium, physical decline, fall, frailty screening, medical review, geriatric assessment, stimulation of cognitive and physical activities. | Educational session, on the job coaching. |
| 4. | Boltz et al. (2014) | Yes (Nurses). | Prevention and management of delirium and functional decline; assessment of mood, cognition, and function; incorporation of function-focused care into care routines; ways to communicate, partnering, and motivating patients and family; and discharge planning. | Education session. |
| 5. | Bryant et al. (2019) | Yes (nurses, ancillary staff, residents, physician assistants and attending staff who frequently care for older patients in trauma unit and in the ICU). | Delirium prevention, mobility, medication management/prognostic assessment, reducing other complications, social needs/goal setting and injury prevention. | Educational presentation, pocket card, mandatory educational modules. |
| 6. | Chang et al (2007) | Yes (nurses). | Mobility. | In-service. |
| 7. | Cohen et al (2019) | Yes (Nurses, nurse aides). | Mobility. | Theoretical and practical training. |
| 8. | Gazineo et al. (2021) | Yes (only one nurse). | Mobility. | Training. |
| 9. | Gill et al (2023) | Yes (unit staff including nurses, nurse aides and physical therapists). | Mobility. | Theoretical and practical training. |
| 10. | Hamilton & Lyon (1995) | Yes (nurses). | Mobility. | Staff education. |
| 11. | Heim et al. (2016). | Yes (nurses and medical staff). | Frailty, ADL functioning. | On the job training. |
| 12. | Holt et al. (2013) | Yes (ward staff including staff nurses and healthcare assistants). | Delirium. | Interactive lecture, handout, quiz, poster, reference materials and case vignettes. |
| 13 | Inouye et al. (1993b) | Yes (Nurses). | Geriatric care (Delirium, functional impairment, incontinence and pressure sores). | Education, training, lecture series. |
| 14. | Inouye et al. (1999) | Yes (Research and clinical nurses). | Delirium risk factors: cognitive impairment, sleep deprivation, immobility, visual impairment, hearing impairment and dehydration. | Training. |
| 15. | Inouye et al. (2000) | Yes (Nurses and Physicians) and volunteers. | Mobility, vision/hearing, volume repletion, feeding assistance, sleep. | Formal didactic sessions, small group training, and resource materials. |
| 16. | Inouye et al. (2006) | Unsure (volunteers). | Delirium. | Regular education sessions |
| 17. | Juneau et al. (2018) | Yes (Nurses, physicians and physiotherapist). | Mobility. | Training on exercise program. |
| 18. | King et al. (2016). | Yes (Nurses and certified nursing assistants). | Mobility. | Training. |

*(Continued)*

**Table 1.** (Continued)

| S.N | First author (Year) | HCPs included in education | Topics covered. (e.g., delirium, mobility) | Education strategies used (e.g., in-service, workshops, pockets cards) |
|---|---|---|---|---|
| 19. | Kratz et al. (2015) | Yes (Nurses). | Delirium. | Training. |
| 20. | Liu et al. (2018) | Yes (Physicians, nurses). | Mobility. | Education modules. |
| 21. | Martinez- Velila et al. (2016) | Yes (ward and research team staff). | Mobility and functional independence. | Education. |
| 22. | Milisen et al. (2001). | Yes (Nurses). | Delirium, depression, and dementia. | Educational poster, training |
| 23. | Miller et al. (2004) | Yes (Nurses and undergraduate nursing students). | Confusion, discomfort, physical care, assessment of basic needs, patient safety and therapeutic communications. | Education program. |
| 24. | Mudge et al. (2008) | Yes (Nurses). | Functional independence and mobility | Education, group in-services |
| 25. | Mudge et al. (2022). | Yes (Nurses or allied health professional). | Nutrition and hydration, mobility, meaningful cognitive and social engagement and in multidisciplinary teamwork. | Training, videos. |
| 26. | Mudge et al. (2020). | Yes (allied health assistants). | Nutrition and hydration, mobility, meaningful cognitive and social engagement and in multidisciplinary teamwork. | Training. |
| 27. | Mudge et al. (2023) | Yes (allied health or nursing assistants). | Nutrition and hydration, mobility, meaningful cognitive and social engagement and in multidisciplinary teamwork. | Training including didactic and interactive content. |
| 28. | Naylor et al. (2014). | Yes (nurses and advanced practice nurses). | Cognitive deficit. | Web based modules, seminars. |
| 29. | Peyrusque et al. (2021) | Yes (nurses, physicians, physiotherapists). | Mobility. | Group training. |
| 30. | Rodrigues et al. (2020). | Yes (nurses and nursing assistants). | Mobility, nutrition. | Training and regular education sessions. |
| 31. | Rubin et al. (2011) | Unsure (Volunteers). | Nutrition, delirium, overall care of frail older adults. | Training program. |
| 32. | Suwanpasu et al. (2015) | Yes (nurses). | Sleep disorder, problem with eating or feeding, incontinence, confusion, mobility, falls, and skin breakdown. | Training program. |
| 33. | Vidan et al. (2009) | Yes (nurses). | Delirium. | Educational session, poster, cards with recommendations included in the first page of the treatment book. |
| 34. | von Renteln-Kruse & Krause (2007). | Yes (Nurses, therapeutic staff, and physicians). | Mobility. | Presentations, training. |
| 35. | Wand et al. (2014). | Yes (nurses, medical staff). | Delirium. | Ward based in-services, workshops, tutorials, education session, posters. |
| 36. | Wang et al. (2020) | Yes (Medical postgraduates, majoring in geriatrics or geriatric nursing were trained as coordinators, who took charge of delivering the intervention plan to nurses, surgical doctors and other team staff). | Delirium. | Training. |
| 37. | Wanich et al. (1992) | Yes (Nurses). | Delirium. | In-service program. |
| 38. | Zisberg et al. (2018). | Yes (Nurses, nurse aides). | Mobility. | Online tutorials, face to face training. |

**Table 2. Interventions in Addition to Education.**

| Study | Level of intervention | | | Care process changes (Nurse only (N); Interprofessional (IP)) | | | | | | | | | | | | Environment changes | | | | | |
|---|---|---|---|---|---|---|---|---|---|---|---|---|---|---|---|---|---|---|---|---|---|
| | Large | Medium * | Small | Multifocal geriatric protocols | Mobility interventions | Family involvement | Cognitive interventions | Continence interventions | Pain interventions | Medication review/protocol | Pressure ulcer prevention | Nutrition/Dehydration | Sleep protocol | Physical Function/Falls | Geriatric/Frailty screening | Geriatric patient rounds | Volunteers | Walking trails | Orientation (clocks) | Modifications/Declutter | Adjusting seating/bed height |
| Allegri et al. (2022) | | * | | N | | | N | | N | N | | | | | | | | | | | |
| Apolinario et al. (2022) | | * | | IP | | IP | | | | | | | | | * | IP | | | | | |
| Bakker et al. (2014). | | * | | IP | | | * | | | * | | * | * | * | * | * | | | | | |
| Boltz et al. (2014) | | * | | N | | * | | | | | | | | | | | | | * | * | * |
| Bryant et al. (2019) | | * | | IP | * | | * | | | * | | * | | * | * | * | | | | | |
| Chang et al (2007) | | * | | N | | | | | | * | | | | | | | | | | * | |
| Cohen et al. (2019). | | | * | | N | * | | | | | | | | | | | | | * | * | |
| Gazineo et al. (2021) | | | * | | N | * | | | | | | | | | | | | | | | |
| Gill et al (2023) | | | * | IP | | | | | | | | | | | | | | | * | * | |
| Hamilton & Lyon (1995) | | | * | | N | | | N | | | | | | | | | | | | | |
| Heim et al. (2016). | * | | | IP | | | * | | | | | * | | * | * | * | | | | | |
| Holt et al. (2013) | | | * | | | | N | | | | | | | | | | | | | | |
| Inouye et al. (1993b) | | * | | N | | | * | * | | | * | * | * | * | | | | | | | |
| Inouye et al. (1999) | | * | | IP | * | | * | | | | | * | * | | | | * | | * | * | |
| Inouye et al. (2000) | | * | | IP | * | * | * | | | | | * | * | | | * | * | | * | * | |

| Administrative Support | | | | | | | Facilitation/Leadership | | | | | | Discharge/ Transition Planning | | | | Functional outcomes improvement |
|---|---|---|---|---|---|---|---|---|---|---|---|---|---|---|---|---|---|
| Sensory and mobility aids | Equipment purchase | Additional staffing | Restraint use/policy | Patient/family in shift report | Electronic record modifications | Regional Intersectoral | Advanced practice nurse | Geriatric resource nurse/staff nurses with addition education | Nurse supervisor/case manager | Geriatrician | IP resource person | IP geriatric team IP geriatric | IP family meetings | Links to discharge settings | Minimum data set/digital info | Patient follow up up | |
| | | | | | | | | | | | | | | | | | **Physical** Maintained **Cognitive** No |
| | | | | | | | | | * | | | | | * | | | **Physical: ADL – Yes** |
| | | | | | | | * | | * | | | | * | | | * | **Physical** ADL No **Cognitive** No |
| | | | * | | | | | * | * | | | | | | | * | **Physical: ADL/Mobility** Yes **Physical:** Gait and Balance No **Cognitive:** Delirium severity Yes |
| | | | | | | | | * | | | | | * | | | | **Cognitive:** delirium Yes |
| | | | | | | | | * | | * | | | | | | | **Physical: ADL Yes** |
| | * | | | * | | | | | | | | | | | | | **Physical:** BADL No IADL No |
| | | | | | | | | * | | | | | | | | | **Physical: Mobility Yes** |
| | * | | | | | | | | | | | | | | | | **Physical: Mobility Yes** |
| | | * | | | | | * | | * | | | | | | | | **Physical: ADL Yes** |
| | | | | * | | | | * | | | | | | * | * | | **Physical: ADL Yes** (composite outcome) |
| | | | | | | | * | * | * | * | | | | | | | **Cognitive –** Delirium Yes |
| | | | | | | | * | * | | * | | | | | | | **Physical: ADL Yes** |
| | * | | * | | | | * | | | * | * | * | | | | | **Physical** ADL- yes (trend) **Cognitive** Delirium Yes |
| | | | | | | | * | | * | * | * | | | * | * | * | **Physical** ADL Yes **Cognitive** Delirium Yes |

*(Continued)*

 

**Table 2.** (Continued)

| Study | Level of intervention | | | Care process changes (Nurse only (N); Interprofessional (IP)) | | | | | | | | | | | | Environment changes | | | | | |
|---|---|---|---|---|---|---|---|---|---|---|---|---|---|---|---|---|---|---|---|---|---|
| | Large | Medium* | Small | Multifocal geriatric protocols | Mobility interventions | Family involvement | Cognitive interventions | Continence interventions | Pain interventions | Medication review/protocol | Pressure ulcer prevention | Nutrition/Dehydration | Sleep protocol | Physical Function/Falls | Geriatric/Frailty screening | Geriatric patient rounds | Volunteers | Walking trails | Orientation (clocks) | Modifications/Declutter | Adjusting seating/bed height |
| Inouye et al. (2006) | * | | | IP | * | | * | | | | | * | * | | | | * | | * | * | |
| Juneau et al. (2018) | | | * | IP | | * | | | | | | | | | | | | | | | |
| King et al. (2016). | | | * | N | | | | | | | | | | | | | | * | * | | |
| Kratz et al. (2015) | | * | | N | * | | * | | | | | * | * | | | | | | * | | |
| Liu et al. (2018) | | * | | IP | | | | | | | | | | | | | | | | | |
| Martinez- Velila et al. (2016) | | | * | IP | | * | | | | | | | | | | | | | | | |
| Milisen et al. (2001). | | * | | N | | | * | | * | * | | | | | | | | | | | |
| Miller et al. (2004) | | * | | N | | * | * | | * | | | | | | | | | | | * | |
| Mudge et al. (2008) | | * | | IP | * | | * | | | | | | | | | * | | * | | | |
| Mudge et al. (2020). | | * | | IP | * | | * | | | | | * | | | | * | | * | * | | |
| Mudge et al. (2022). | * | | | IP | * | | * | | | | | * | | | | | | * | * | | |
| Mudge et al. (2023) | * | | | IP | * | | * | | | | | * | | | | | | * | * | | |

| Administrative Support | | | | | | | Facilitation/Leadership | | | | | | Discharge/ Transition Planning | | | | Functional outcomes improvement |
|---|---|---|---|---|---|---|---|---|---|---|---|---|---|---|---|---|---|
| Sensory and mobility aids | Equipment purchase | Additional staffing | Restraint use/policy | Patient/family in shift report | Electronic record modifications | Regional Intersectoral | Advanced practice nurse | Geriatric resource nurse/staff nurses with addition education | Nurse supervisor/case manager | Geriatrician | IP resource person | IP geriatric team IP geriatric | IP family meetings | Links to discharge settings | Minimum data set/digital info | Patient follow up up | |
| | * | | * | | | | * | | | | * | * | | | | * | **Physical** ADL Yes **Cognitive** Delirium Yes |
| | * | | | | | | | | | | | * | | | | | **Physical** ADL maintained. |
| | * | | | | | | | | | | | * | | | | | **Physical** Mobility Yes |
| | | | | | | | | * | | | | | | | | | **Cognitive** Delirium Yes |
| | | | | | | | | * | | * | | | | | | | **Physical** Mobility Yes |
| | | | | | | | | | | | | | | | | * | **Cognitive** Delirium Yes |
| | | | | | | | * | * | | * | | | | | | | **Cognitive** Delirium Yes |
| | | | | | | | * | | * | | | | | | | | **Physical** ADL No |
| | | | | | | | | * | | | | | | | | | **Physical** ADL Yes **Cognitive** Delirium Yes |
| | | | | | | | | | | | | | * | | | | **Physical** Yes **Cognitive** Delirium Yes |
| | * | * | | | | | | | | | | * | | | | | **Physical** No (composite score) **Cognitive** Delirium Yes |
| * | | * | | | | | | | | | | * | | | | | **Physical** Yes **Cognitive/ social** Yes |

*(Continued)*

| Study | Level of intervention | | | Care process changes (Nurse only (N); Interprofessional (IP)) | | | | | | | | | | | | Environment changes | | | | | |
|---|---|---|---|---|---|---|---|---|---|---|---|---|---|---|---|---|---|---|---|---|---|
| | Large | Medium* | Small | Multifocal geriatric protocols | Mobility interventions | Family involvement | Cognitive interventions | Continence interventions | Pain interventions | Medication review/protocol | Pressure ulcer prevention | Nutrition/Dehydration | Sleep protocol | Physical Function/Falls | Geriatric/Frailty screening | Geriatric patient rounds | Volunteers | Walking trails | Orientation (clocks) | Modifications/Declutter | Adjusting seating/bed height |
| Naylor et al. (2014). | | | * | | | | N | | | | | | | | | | | | | | |
| Peyrusque et al. (2021) | | | * | | IP | * | | | | | | | | | | | | | | | |
| Rodrigues et al. (2020). | | * | | N | * | | * | | | | | * | | | | | | * | * | * | |
| Rubin et al. (2011) | | * | | IP | * | * | * | | | | | * | * | | | | | | | * | |
| Suwanpasu et al. (2015) | | * | | N | * | | * | * | | | | * | * | * | | N | | | | | |
| Vidan et al. (2009) | | * | | N | * | | * | | * | | | * | | * | | | | | * | | |
| von Renteln-Kruse & Krause (2007). | | | * | | | * | | | * | | | | | IP | | | | | | | |
| Wand et al. (2014) | | * | | IP | | | * | * | * | | | * | * | | | | | | * | | |
| Wang et al. (2020) | | * | | IP | * | * | * | | * | | | * | * | | | * | | | | | |
| Wanich et al. (1992) | | * | | N | * | * | * | | | * | | | | | | | | | * | | |
| Zisberg et al. (2018). | | | * | | IP | * | | | | | | | | | | | | | * | | |

*Note:* N = Nurse; IP = Interprofessional; ADL = activities of daily living; IADL = instrumental activities of daily livi.

| Administrative Support | | | | | | | Facilitation/Leadership | | | | | | Discharge/ Transition Planning | | | | Functional outcomes improvement |
|---|---|---|---|---|---|---|---|---|---|---|---|---|---|---|---|---|---|
| Sensory and mobility aids | Equipment purchase | Additional staffing | Restraint use/policy | Patient/family in shift report | Electronic record modifications | Regional Intersectoral | Advanced practice nurse | Geriatric resource nurse/staff nurses with addition education | Nurse supervisor/case manager | Geriatrician | IP resource person | IP geriatric team IP geriatric | IP family meetings | Links to discharge settings | Minimum data set/digital info | Patient follow up up | |
| | | | | | | | | * | | | | | | | | * | **Physical** ADL/IADL No |
| | * | | | | | | | | | | | * | | | | | **Physical** ADL, Mobility Yes |
| | | | | | | | | | | | | | | | | | **Physical** ADL Yes |
| | | | | | | | * | * | | * | | * | | | | | **Cognitive** Delirium Yes |
| | | | | | | | * | | | | | * | | | | | **Physical** ADL No |
| | | | * | | | | * | | * | | | | | | | | **Physical** ADL Yes **Cognitive** Delirium Yes |
| * | | | | | | | | | | | | | | | | | **Physical** Falls risk Yes |
| | | | * | | | | | * | * | | | | | | | | **Physical** ADL Yes **Cognitive** Delirium Yes |
| | | | | | | | | | | | | * | | | | | **Physical** ADL Yes **Cognitive** Delirium Yes |
| * | | | | | | | * | | | | | | | | | | **Physical** ADL Yes |
| | * | | | | * | | | | * | | | | | | | | **Physical** Mobility yes |

**Administration/management support.** Administrative/management support included provision of equipment [38,40,56,61,64,68–71], restraint policies [40,45,60,71], electronic record modifications [38,70], and additional staffing [49,64,65]. Only one study involved regional inter-sectoral collaboration [34].

**Leadership support/facilitation.** Facilitation/leadership roles in practice change included advanced practice nurses [36,40,45,46,49–52,55,58,59,62,71], geriatric resource nurse/staff [37,39,42,43,47,48,50,51,53–55,58,60,63], and nurse supervisor/case managers [35,47,49,50,52,62,70]. Additional personnel included geriatricians [36,40,45,48,50–55,60,71], interprofessional resource persons [40,52,58,68,69,71], and interprofessional geriatric teams [40,46,56,58,64,65,67].

**Discharge/transition planning.** Discharge/transition planning included interprofessional family meetings [36,37,44], links to discharge settings [35,39,52], and minimum data set information [39,52]. The most common discharge/transition process involved patient follow-up [36,41,42,47,52,71].

## Measured outcomes

Most studies reported on change in physical function (baseline to discharge) [35,38,46,48,49,51,52,54,56,57,59,61–63,66,68–70]. Only a few studies used self-report [38,66] or caregiver report [42] of pre-hospital function as baseline; the remainder were performance-based assessments by healthcare professionals. Six studies reported on cognitive function, primarily delirium [37,41,50,53,55,58], and 12 reported on both physical and cognitive outcomes [34,36,40,43–45,47,52,60,65,67,71]. All studies included nurse or interprofessional multifocal protocols. Two studies reported on a composite score: one an adverse outcome score measuring physical function and high healthcare demand post-hospital [39] and the other a composite of hospitalized outcomes such as delirium and pressure injury [64]. The major physical function domains assessed were related to mobility, dexterity, and ADL. The major cognitive function domains assessed included orientation to time and place, recall etc. The most common measures included the Functional Independence Measure, Barthel Index, Katz Activities of Daily Living Scale, Mini Mental State Exam (MMSE), and the Montreal Cognitive Assessment (MoCA).

## Discussion

In addition to education, a variety of interventions supported implementation of knowledge from the educational initiatives to improve functional outcomes for hospitalized older adults: changes to care processes (e.g., multifocal protocols, mobility interventions, and cognitive interventions), changes to the built environment (e.g., walking trails, clocks, decluttering), administration/management support (e.g., purchase of equipment, electronic record modification), leadership support/facilitation (e.g., geriatricians and specialized nurses), and discharge/transition planning. We used a tripartite conceptualization to describe the scale of the interventions. Half the studies (18/38) concentrated on nurses for education and the delivery of the intervention. This could be because nurses make up almost half of the global health workforce [72] and are the healthcare professionals that interact the most with patients in hospitals [73]. In recent years, there has been increasing recognition of the importance of interprofessional collaboration in complex care [14]. The trend from uni-professional interventions to multi-professional interventions was observed in our findings; studies published in 2013 or earlier [40,43,48,51,59,62] tended to focus exclusively on nurse-driven interventions, whereas studies published in the last decade (e.g., [34–37,39,44,54,56,61,64,67,71]) typically incorporated an interprofessional approach. This typically entails multiple healthcare workers from different professional backgrounds working together with patients, families, carers and communities to deliver the highest possible quality care across settings [74]. The professionals involved differ on a case-by-case basis, but in the acute care context, this would normally involve physicians, nurses, and allied health professionals (e.g., physiotherapists, occupational therapists, dieticians etc.).

The studies reported that educational interventions to enhance knowledge uptake had a measurable effect on functional outcomes that were overwhelmingly physical and/or cognitive, with very few studies focusing on cognitive function exclusively. Cognitive function is now receiving more scholarly attention [75] due to increasing recognition that cognitive

and physical function are interconnected [76]. However, our findings echo those of a recent systematic review3 which found that cognition is largely ignored; of 94 included studies, 93 focused exclusively on physical deconditioning.

Our review also highlights gaps in the existing research in terms of theory, research designs, intervention descriptions, country representation, policy, and environmental changes. Included studies typically reported on time-limited initiatives and did not address sustainability and how the intervention would be maintained after the study [77]. The WALK-FOR program is an exception as it was developed and tested on two units [38,70] and examined the sustainability in a third study [61]. Another model, EAT WALK ENGAGE, was tested for effectiveness on a single unit and then expanded to eight units in four hospitals [64,65], but was not examined for sustainability. Notably, the HELP model [51] spread to other hospital sites and was assessed for sustainability [71]. Moreover, elements of the HELP model were incorporated into other studies [67]. Future reviews should examine the characteristics of these models/programs that enhance sustainability.

Another gap is the absence of theoretical frameworks – only three studies explicitly used a structured framework to guide development/implementation of their intervention model [64,65,70], likely due to studies pre-dating recent advances in implementation science. Moreover, many included studies seemed to view change as a linear process, although current theoretical work emphasizes complexity and construes change as non-linear [33,78]. As such, some included studies failed to account for contextual elements previously identified as barriers to older adults' care (e.g., short staffing, lack of resources, a medical focus, inter-professional team inaccessibility during evenings and weekends) [14,79–83]. We suggest that future work in this area be guided by implementation theory.

Other identified gaps include a lack of rigorous designs, descriptions of interventions, education components, and representation. Only five of the 38 studies used a randomized controlled design. We are conscious that randomized controlled trials (RCTs) – typically viewed as the gold standard in clinical research – are not always the most appropriate method for every research question. RCTs can be undermined by high costs, ethical considerations, and insufficient generalizability to broader, more diverse populations. We therefore believe that more rigorous studies on the topic are needed. Such studies should include RCTs as well as qualitative and mixed methods approaches to understand the complexity of practice change. Descriptions of interventions were inconsistently presented, making it difficult to understand the education intervention and other initiatives provided to support implementation. A large number of studies were excluded during the review process because the education element was not described.

Included studies generally did not concentrate on policy and changes to the built environment, despite evidence that the built environment can contribute to improved functioning in older adults [84]. Most of the included environmental changes were minor/superficial (e.g., clocks, decluttering). No studies described a major renovation to the built environment, possibly because the cost is prohibitive. There was little evidence on administrative changes to support interventions; a few studies describing restraint policies or practices were identified [40,45,60,71]. Most administrative support was minor except for Heim et al. [39], in which a system change was undertaken.

## Strengths and limitations

This study adds to the literature on care for hospitalized older adults by synthesizing the evidence on education that supported intervention implementation, scaling/spread, and sustainability. The study also provides a close examination of other interventions supporting the implementation of gerontological knowledge, which are typically overlooked in reviews. We also grounded our study in a robust, context-sensitive theoretical conceptualization drawn from ecological implementation science frameworks. We demonstrated that prior initiatives are mostly local in scope and piece-meal, with very few initiatives cutting across a healthcare system. We searched the literature in other languages, thereby accessing valuable studies that would have been excluded in an English-only review. However, we did not search the grey literature, dissertations/theses, or Clinical Practice Guidelines. Moreover, it is possible that education may have been delivered to healthcare providers in some studies that were not reported on (and therefore those studies were mistakenly excluded when assessing eligibility). We also focused our search on physical function and did not include cognitive interventions or other

environmental interventions in our search terms (though we did extract any cognitive outcomes reported). Future reviews should explicitly address cognitive function. Additionally, we adhered to the definition of older adults as aged 65 + used in many World Health Organization reports [31], admittedly an arbitrary cutoff point, and we thus excluded studies which defined older adults as 64 years of age or younger that may have otherwise been eligible. Finally, as with all scoping reviews, there is also an unknown risk of publication bias due to the tendency of researchers to publish positive, statistically significant findings, and to not publish statistically insignificant or negative results. Finally, most studies were conducted in high-income countries and their findings may not be applicable to low- and middle-income countries.

## Directions for future research and policy

This scoping review, which mapped the breadth of the existing evidence, lays the foundation for future systematic reviews [33]. Building on this study, future systematic reviews should identify which of the interventions we identified are most effective in optimizing the functioning of hospitalized older adults, and by comparing single interventions with multi-intervention programs. Scholars should then develop strategies based on implementation science theory to ensure successful implementation of the most effective interventions while taking into consideration the realities of hospital units, such as short staffing, resource constraints, and overcrowding. Researchers reporting on similar interventions should describe the education component of interventions to enhance transparency and facilitate replication. Further work needs to be undertaken to integrate care of older persons across healthcare systems, and policy-level changes are required to facilitate the provisioning of function-oriented care.

## Supporting information

**S1 Table. Preferred Reporting Items for Systematic reviews and Meta-Analyses extension for Scoping Reviews (PRISMA-ScR) Checklist.**
(DOCX)

**S2 Table. Inclusion and Exclusion Criteria.**
(DOCX)

**S3 Table. Search Strategies for MEDLINE.**
(DOCX)

**S4 Table. Table of Included Studies (N = 38).**
(DOCX)

**S1 Fig. Preferred Reporting Items for Systematic Reviews and Meta-Analyses (PRISMA) Flow Diagram.**
(DOCX)

## Author contributions

**Conceptualization:** Kathleen F. Hunter, Adrian Wagg.

**Formal analysis:** Kathleen F. Hunter, Jeffrey I. Butler, Shovana Shrestha, Nick Millar, Sherry Dahlke.

**Funding acquisition:** Kathleen F. Hunter, Adrian Wagg.

**Methodology:** Kathleen F. Hunter.

**Project administration:** Kathleen F. Hunter.

**Software:** Kathleen F. Hunter.

**Supervision:** Kathleen F. Hunter.

**Validation:** Kathleen F. Hunter, Jeffrey I. Butler.

**Writing – original draft:** Kathleen F. Hunter, Jeffrey I. Butler, Sherry Dahlke.

**Writing – review & editing:** Kathleen F. Hunter, Jeffrey I. Butler, Shovana Shrestha, Adrian Wagg, Frances Carr, Sherry Dahlke, C. Allyson Jones.

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
