## [Decision Letter · Decision Letter 0]

PONE-D-24-35120Interventions Supporting Implementation of Gerontological Education for Healthcare Professionals to Improve Functional Outcomes for Hospitalized Older Adults: A Scoping ReviewPLOS ONE

Dear Dr. Hunter,

Thank you for submitting your manuscript to PLOS ONE. After careful consideration, we feel that it has merit but does not fully meet PLOS ONE’s publication criteria as it currently stands. Therefore, we invite you to submit a revised version of the manuscript that addresses the points raised during the review process.

We look forward to receiving your revised manuscript.

Kind regards,

Mohammad Jamil Rababa

Academic Editor

PLOS ONE

Journal Requirements:

2. We notice that your supplementary [figures/tables] are included in the manuscript file. Please remove them and upload them with the file type 'Supporting Information'. Please ensure that each Supporting Information file has a legend listed in the manuscript after the references list.

3. We note you have included a table to which you do not refer in the text of your manuscript. Please ensure that you refer to Table 4 in your text; if accepted, production will need this reference to link the reader to the Table.

Reviewers' comments:

Reviewer's Responses to Questions

**Comments to the Author**

1. Is the manuscript technically sound, and do the data support the conclusions?

Reviewer #1: Partly

Reviewer #2: Yes

2. Has the statistical analysis been performed appropriately and rigorously? 

Reviewer #1: Yes

Reviewer #2: N/A

3. Have the authors made all data underlying the findings in their manuscript fully available?

Reviewer #1: Yes

Reviewer #2: Yes

4. Is the manuscript presented in an intelligible fashion and written in standard English?

Reviewer #1: Yes

Reviewer #2: Yes

5. Review Comments to the Author

Reviewer #1: The paper addresses the important questions of how acute hospitals can maintain the physical and cognitive functions of elderly inpatients. While the paper categorized various such interventions, the authors are suggested to consider the following points:

1. The paper chooses to review the papers that contain the element of education for staff. In addition to knowledge deficit, there are numerous factors of why there were insufficient interventions that aimed at physical and cognitive functions. No sufficient reasons and theory support were described to backup the choice of focusing on education. While a typical staff training that aims at improving functional outcomes usually involves staff education, such choice led to authors excluding papers that did not elaborate staff knowledge training. More justifications are needed to support this choice.

2. More quantitative findings could be cited in the introduction section to show the adverse effects of acute hospital in worsening physical and cognitive function, and thereby highlight the importance of the paper.

3. The research priority mentioned on line 94 can be moved to the introduction section.

4. More elaborations of the terms in the introduction section are desirable, e.g. “ecological concepts from implementation science” on line 62, “numerous factors on line 80”.

5. As per my understanding, acute hospital focuses on short-term medical treatment. For each study, for how long do the participants stay in the hospital and receive the interventions? I wonder if the focus of acute hospital interventions should be “improve” or maintain functional outcomes as stated on line 107 due to the short-term nature of hospital stay.

6. On line 122-124, “The search focused on physical function and BADL, but both physical and cognitive outcomes were abstracted from any included studies”. Please explain this in the introduction section.

7. What are the major physical and cognitive function domains and measures among the papers reviewed?

8. The intervention categories did not seem to be discussed in detail in discussion section. What is “variety of interventions” as on line 243? Please explain in sufficient detail to show that this is the objective of the paper as stated in the Objective section.

9. Nurses as primary interventionist vs “Interprofessional approach” is the focus in the first paragraph in discussion section. What are the professions in “Interprofessional approach”? How are the division of roles and cooperation among the professions in the papers reviewed?

10. The third paragraph in the discussion section concerns sustainability but elaborate if some interventions were expanded to other hospitals. Please define sustainability.

Reviewer #2: This ScR addresses a relevant topic as functional decline is a common problem among hospitalized older people especially for long stay hospitalization. Authors followed conduct and reporting guidelines specific for Scoping reviews. Methods were well described and the review results answer the set objectives and highlight reserach gaps.

Review comments

Title

- To simplify/clarify the title: “Interventions Supporting Implementation of Gerontological Education for Healthcare Professionals to Improve Functional Outcomes for Hospitalized Older Adults: A Scoping Review”; could be: interventions supporting gerontological education for healthcare professionals to improve … OR interventions supporting the translation of gerontological knowledge into practice to improve functional outcomes for … OR Interventions Supporting the Implementation of healthcare professionals’ Gerontological knowledge to Improve

Abstract:

- L.30: examine what? Authors may want to add “the characteristics or type of interventions”

- L40: Interventions included in addition to educational strategies, care processes …

- L40: environmental changes and leadership roles.

- Add relevant key words to facilitate retrieval of the study (may include healthcare professionals, implementation science)

- Add source of evidence to the abstract (as per Prisma)

- Words describing the main examined concept should be unified and maintained throughout the manuscript starting with the title and especially when talking about aims and objectives in different sections to avoid misleading the readers. When I first read the manuscript, I wasn’t sure whether the focus was merely on educational interventions or other supportive strategies to implement knowledge and improve functional outcomes. It is until the results section that things got clear. It could be “initiatives supporting educational interventions for healthcare professionals to improve functional outcomes among hospitalized older adults” OR as per the title “interventions supporting gerontological education for healthcare professionals to improve…” RATHER THAN “interventions that have been used to support the implementation of gerontological knowledge by hospital-based healthcare professionals and/or interprofessional teams to improve older adults’ functioning” (in the abstract)

- L34: “Target studies examined initiatives or care delivery models to improve functional outcomes”: this could be removed and keep in L35: “Included studies described an educational intervention for healthcare professionals as part of a wider initiative to promote practice change and assessed functional outcomes (physical and/or cognitive)”.

Introduction:

- Justify the use of the ScR approach; why authors did not choose other approaches? As per Prisma checklist and Peters’ et al., the objectives of this review should be aligned with those of the ScR approach: “Explain why the review questions/objectives lend themselves to a scoping review approach”. (for instance: explore the breadth or depth of the literature, map and summarize the evidence, inform future research, and identify or address knowledge gaps, heterogeneous literature, precursor of systematic review, identify key characteristics or factors related to the concept).

Methods

- L92 “We used the steps for scoping reviews recommended by Peters et al.” better to say guidelines as this reference does not include steps. It rather clarifies guidelines related to ScR conduct, in line with PRISMA-ScR.

- L93 “Our project sought to understand how healthcare professionals’ gerontological knowledge can be improved and applied, one of the research priorities identified in a provincial priority-setting partnershi”: the wording of the ScR aim should be consistent. While it was in the aim section, “to determine what interventions for hospital-based healthcare professionals and/or interprofessional teams, in addition to gerontological education, have been implemented to optimize older adults’ functional outcomes in hospital settings”, here it turned to be “how healthcare professionals’ gerontological knowledge can be improved and applied”. Those are two different research questions.

- Line 95-102: justification for the use of ScR approach: I suggest this is removed to the introduction section, to anchor the choice of the approach in the available literature, what do we know about the topic and what could this review add. Align the objective of this study with the objectives of the ScR itself. How could ScR help to meet the study aims?

- L104: refer to the availability of the checklist as supplementary file.

- Inclusion in L106 (all designs) contradicts L11 (excluded reviews): add to L106 all designs of primary research

- L106: there is a need to explain how studies with different languages were handled (translation? Multilingual authors?)

- L112 “less than 65” justify here the cutoff age (used by WHO as explained in later sections)

- L113: specify (supplementary table 3)

- L119: search strategy: add here the main examined concept: “educational interventions…”. “intervention” as a concept was mentioned in the Medline strategy; however, the population (PCC mnemonic) was absent (health care professionals-interprofessional teams). The cognitive function was also absent but authors mentioned this as a limitation.

- L115-139: the review is not blinded so pay attention to add authors initials.

Results

- L185: suggest to replace "capture” with “characterize”

- L188: managerial intervention, leadership or facilitation roles

- L201: medium scale interventions and not studies

- L201: add “on” after focused

- Table 1 include details about the studies and is lengthy; authors may decide to include a summarized table in the text and add this one as supplementary file.

Discussion

- Strengths: “We also grounded our study in a robust theoretical conceptualization”: not explained-registration link not accessible

- L309: referring to 65+ as a WHO definition of older age is not accurate, even in the cited reference. UN defines older people as aged over 60. WHO use 60+ in some reports and 65+ in others. Better to say 65+ as a cutoff age used in many WHO reports.

Supplementary files:

- At the end of references add all supplementary files (search strategy, inclusion criteria table, and PRISMA flow diagram in addition to the PRISMA checklist)

- Prisma flow diagram:

o reverse first arrow to the right.

o If 4482 are removed at the level of title and abstract screening, what are the 3921 removed first? Are they duplicates?

o The box of excluded studies (screening level) is incomplete. The rest might include other exclusion causes like not describing the education intervention (as mentioned earlier in the text).

6. PLOS authors have the option to publish the peer review history of their article (what does this mean? ). If published, this will include your full peer review and any attached files.

**Do you want your identity to be public for this peer review?** For information about this choice, including consent withdrawal, please see our Privacy Policy .

Reviewer #1: No

Reviewer #2: No

---

## [Author Response · Author response to Decision Letter 1]

24 Jan 2025

Hello, I received a request to confirm that our submission contains all raw data required to replicate the results of your study. Please note that this is correct and that data needed to replicate the study is included in the supplementary files.

Regards,

Kathleen Hunter

---

## [Decision Letter · Decision Letter 1]

PONE-D-24-35120R1Interventions Supporting the Translation of Gerontological Evidence into Practice to Optimize Functional Outcomes for Hospitalized Older Adults: A Scoping ReviewPLOS ONE

Dear Dr. Hunter,

Thank you for submitting your manuscript to PLOS ONE. After careful consideration, we feel that it has merit but does not fully meet PLOS ONE’s publication criteria as it currently stands. Therefore, we invite you to submit a revised version of the manuscript that addresses the points raised during the review process.

We look forward to receiving your revised manuscript.

Kind regards,

Mohammad Jamil Rababa

Academic Editor

PLOS ONE

Journal Requirements:

Reviewers' comments:

Reviewer's Responses to Questions

**Comments to the Author**

1. If the authors have adequately addressed your comments raised in a previous round of review and you feel that this manuscript is now acceptable for publication, you may indicate that here to bypass the “Comments to the Author” section, enter your conflict of interest statement in the “Confidential to Editor” section, and submit your "Accept" recommendation.

Reviewer #1: All comments have been addressed

Reviewer #2: All comments have been addressed

2. Is the manuscript technically sound, and do the data support the conclusions?

Reviewer #1: Yes

Reviewer #2: Yes

3. Has the statistical analysis been performed appropriately and rigorously? 

Reviewer #1: Yes

Reviewer #2: N/A

4. Have the authors made all data underlying the findings in their manuscript fully available?

Reviewer #1: Yes

Reviewer #2: Yes

5. Is the manuscript presented in an intelligible fashion and written in standard English?

Reviewer #1: Yes

Reviewer #2: No

6. Review Comments to the Author

Reviewer #1: The authors have properly addressed my concerns. The introduction and discussion section now seem coherent to me.

Reviewer #2: The manuscript technically reads better after addressing reviewers' comments. However, proofreading is needed to enhance its quality and clarity(rephrasing, syntax, etc). Spotted ambiguous sentences and mistakes are highlighted in the attached file that include my comments. Specific attention is required to the strenghts and limitations section; in addition to syntax issues, the discussion related to generalizibility and positive results within the scoping review approach requires reconsideration.

7. PLOS authors have the option to publish the peer review history of their article (what does this mean? ). If published, this will include your full peer review and any attached files.

**Do you want your identity to be public for this peer review?** For information about this choice, including consent withdrawal, please see our Privacy Policy .

Reviewer #1: No

Reviewer #2: No

---

## [Author Response · Author response to Decision Letter 2]

11 Mar 2025

Please see attached response document.

---

## [Decision Letter · Decision Letter 2]

PONE-D-24-35120R2Interventions Supporting the Translation of Gerontological Evidence into Practice to Optimize Functional Outcomes for Hospitalized Older Adults: A Scoping ReviewPLOS ONE

Dear Dr. Hunter,

Thank you for submitting your manuscript to PLOS ONE. After careful consideration, we feel that it has merit but does not fully meet PLOS ONE’s publication criteria as it currently stands. Therefore, we invite you to submit a revised version of the manuscript that addresses the points raised during the review process.

Dear Respected Authors,  **Please check on these minor comments raised by reviewer 2: **Reviewer #2: The manuscript reads better after addressing all comments. Good luck on your final submission.

Below, I draw your attention to minor required editing (in the track change version):

- L229: built instead of build

- L309: "identified gaps" instead of "gaps identified"

- L340: were instead of was

We look forward to receiving your revised manuscript.

Kind regards,

Mohammad Jamil Rababa

Academic Editor

PLOS ONE

Journal Requirements:

Reviewers' comments:

Reviewer's Responses to Questions

**Comments to the Author**

1. If the authors have adequately addressed your comments raised in a previous round of review and you feel that this manuscript is now acceptable for publication, you may indicate that here to bypass the “Comments to the Author” section, enter your conflict of interest statement in the “Confidential to Editor” section, and submit your "Accept" recommendation.

Reviewer #2: All comments have been addressed

2. Is the manuscript technically sound, and do the data support the conclusions?

Reviewer #2: Yes

3. Has the statistical analysis been performed appropriately and rigorously? 

Reviewer #2: N/A

4. Have the authors made all data underlying the findings in their manuscript fully available?

Reviewer #2: Yes

5. Is the manuscript presented in an intelligible fashion and written in standard English?

Reviewer #2: Yes

6. Review Comments to the Author

Reviewer #2: The mauscript reads better after addressing all comments. Good luck on your final submission.

Below, I draw your attention to minor required editing (in the track change version):

- L229: built instead of build

- L309: "identified gaps" instead of "gaps identified"

- L340: were instaed of was

7. PLOS authors have the option to publish the peer review history of their article (what does this mean? ). If published, this will include your full peer review and any attached files.

**Do you want your identity to be public for this peer review?** For information about this choice, including consent withdrawal, please see our Privacy Policy .

Reviewer #2: No

---

## [Author Response · Author response to Decision Letter 3]

15 Apr 2025

Please see attached response to reviewers table.

---

## [Editor Report · Decision Letter 3]

Interventions Supporting the Translation of Gerontological Evidence into Practice to Optimize Functional Outcomes for Hospitalized Older Adults: A Scoping Review

PONE-D-24-35120R3

Dear Dr. Hunter,

We’re pleased to inform you that your manuscript has been judged scientifically suitable for publication and will be formally accepted for publication once it meets all outstanding technical requirements.

Kind regards,

Mohammad Jamil Rababa

Academic Editor

PLOS ONE
---

## [Editor Report · Acceptance letter]

PONE-D-24-35120R3

PLOS ONE

Dear Dr. Hunter,

I'm pleased to inform you that your manuscript has been deemed suitable for publication in PLOS ONE. Congratulations! Your manuscript is now being handed over to our production team.

Kind regards,

on behalf of

Dr. Mohammad Jamil Rababa

Academic Editor

PLOS ONE